# Mechanical ordering of pigment crystallites in oil binder: Can EPR reveal the gesture of an artist?

Elise Garel, Laurent Binet and Didier Gourier

Chimie-ParisTech, PSL University, CNRS, Institut de Recherche de Chimie-Paris, F-75005 Paris, France

*Correspondence to*: Laurent Binet (laurent.binet@chimieparistech.psl.eu)

**Abstract.** Is it possible to reconstruct the gesture of an ancient artist applying a paint layer, considering that the orientation distribution of crystallites of an inorganic pigment remains definitively imprinted on the support after drying of the layer? If the pigment contains paramagnetic transition metal ions whose magnetic interactions are themselves anisotropic, then the shape of the EPR spectrum should reflect the distribution of grain orientations. We have demonstrated this effect in the case of Egyptian blue $CaCuSi_4O_{10}$, a pigment used for at least three millennia in antiquity, by reconstructing the probability density of crystallite orientations under various modes of application, such as brush-painting, dabbing and droplet deposition.

## 1. Introduction

In magnetic resonance, including Electron Paramagnetic Resonance (EPR), the interactions contained in the spin Hamiltonian are essentially anisotropic. For a system in a rigid state, much information on the structure at the molecular scale of the observed species can be obtained from the anisotropic parts of these interactions. The anisotropic interactions can also be helpful to analyze the orientation distributions of paramagnetic species in disordered materials, thus providing information on the texturation of the sample at a macroscopic scale (Hentschel, et al., 1978; Friesner, et al., 1979). For instance, EPR has already been used to investigate ordering in liquid crystals (Meirovitch, et al., 1982; Imrie, et al., 1997; Yankova, et al., 2013; Bogdanov and Vorobiev, 2022), in polymer films (Vorobiev and Chumakova, 2005), in muscle fibers (Fajer, 1994), in oriented bacteria (Franck et al., 1979) or of graphene oxide membranes (Chumakova, et al., 2022). This approach should also be fruitful in the field of cultural heritage where complex composite magnetic materials are often to be found in cultural artifacts. As an example, paintings are generally made of layers of pigment grains dispersed in a polymer binder. Most often pigments contain paramagnetic transition ions as coloring species. It can be expected that the orientation of the inorganic grains within the layer is not random but rather keeps the memory of the painter's gesture. Therefore, analyzing the orientation distribution of pigment grains within a painting would potentially provide useful information on the making. In this paper, we aim at testing the potential of EPR in determining the orientation distributions of pigment grains in a polymer film. The chosen system is

cuprorivaite $CaCuSi_4O_{10}$ also known as Egyptian blue, a pigment widely used in the Mediterranean basin since about 2500 BC (4[th] Egyptian dynasty) until the end of the Roman empire, its manufacturing recipe having been lost around the seventh century AD (Pagès-Camagna and Colinart, 2003). In this work we study by EPR the orientation effects of cuprorivaite crystallites dispersed in dried linseed oil binder. Different deposition modes of the mixture on a substrate were tested to investigate their influence on the orientation distribution.

## 2. Experimental procedures

### 2.1. Sample preparation

The Egyptian blue pigment was synthetized by solid-state reaction from calcium carbonate $CaCO_3$, amorphous silica $SiO_2$, copper oxide $CuO$ and 3 w.\% of sodium carbonate $Na_2CO_3$. The powders were mixed and ground together then pressed into pellets under a uniaxial pressure of 4 tons/cm$^2$. The pellets were first sintered in air at 1000 °C for 5h. The resulting samples were then ground, pressed into new pellets, and sintered again during 17 h in air at 1000 °C. After this final sintering, the samples were ground again in powder. The purity of the material was tested by X-ray diffraction (XRD) using an X Panalytical X'Pert Pro diffractometer with the $K_{\alpha 1}$ ray of a copper anticathode ($\lambda = 0.15406$ nm). XRD patterns show that the synthesized powder consists of cuprorivaite $CaCuSi_4O_{10}$ with traces of wollastonite $CaSiO_3$ and $SiO_2$ (Binet et al., 2021). Cuprorivaite is a member of the Phyllosilicate group, all minerals of which have the form of platelets. Its structure is made up of double layers of corner-sharing $[SiO_4]$ tetrahedra, separated by a layer of $Ca^{2+}$ ions. It is tetragonal, with space group $P4/_{nnc}$ (Pabst, 1959), which means that the crystallographic $c$-axis ($C_4$ axis) is normal to the silicate layers, and thus to the platelets. $Cu^{2+}$ ions occupy plane-square sites with $D_{4h}$ point symmetry (Pabst, 1959), with the $C_4$ axis (which is also the g$_{//}$ axis) parallel to the crystallographic $c$-axis.

The as-obtained Egyptian Blue pigment was then mixed with boiled linseed oil as a binder in proportion of 10 w.\% of pigment. The liquid pigment/oil mixture was deposited on horizontal supports to allow drying. Either a glass plate or on a semi-rigid transparent polymer substrate (transparency for overhead projector) were used. Four different deposition methods were tested (Fig. 1): (i) a one-way spreading with a film applicator to produce a 250 μm thick (before drying) film, (ii) a one-way spreading with a paintbrush, (iii) a dabbing with a paintbrush, (iv) a droplet deposition with a pipette. The samples were then let dry for several days until the linseed oil becomes rigid. Of relevance for further discussion is the fact that during the drying, the vertical direction, hence gravity, was perpendicular to the sample plane. For EPR analysis and for samples which had to be rotated about an in-plane direction, small pieces with dimensions 25 mm x 2-3 mm were cut perpendicularly to the spreading direction (when relevant) as shown in Fig. 1. For samples which had to be rotated about the normal to the plane, pieces of approximate size 2 mm x 2 mm were cut to fit into standard EPR tubes.

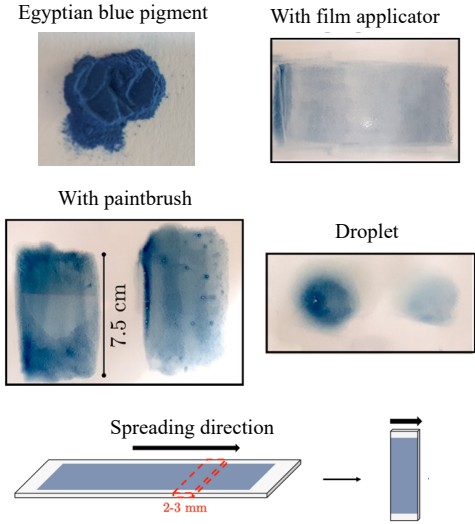

**Figure 1.** Photographs of the samples

2.2. EPR analysis

EPR spectra were recorded at room temperature with a Bruker Elexsys E500 spectrometer equipped with Bruker SHQE
resonator operating at X-band (microwave frequency $\approx$ 9.25-9.39 GHz depending on the sample). The applied magnetic field
was modulated at 100 kHz with a 1 mT amplitude for lock-in detection. The microwave power was 2 mW. To analyze
theoretically the orientation distribution of pigment grains in a fluid medium, a sample made of pigment powder in fluid linseed
oiled was submitted to a 20 mT magnetic field in the EPR spectrometer, at room temperature. Then the sample was cooled
down to 200 K with gaseous nitrogen to freeze the oil and to keep the grain orientations fixed in the sample. EPR spectra at
different orientations of the sample in the external field were then recorded at this temperature.

All calculations for EPR spectra analysis were performed with Matlab (script given in Supplementary Information). The
Easyspin package (Stoll and Schweiger, 2006) was used for EPR spectrum simulation, by taking an axial $g$-matrix with
principal values $g_x = g_y = 2.055 = g_\perp$ and $g_z = 2.350 = g_\parallel$ for $Cu^{2+}$ in the Egyptian Blue pigment. Here the $(x, y, z)$
molecular frame is such as the $z$-axis is the $C_4$ symmetry axis of the $Cu^{2+}$ coordination site with $D_{4h}$ symmetry (Ford and
Hitchman, 1979). This axis is also the crystal $c$-axis of the structure (Pabst, 1959). A Voigt line shape was considered for the
EPR transitions with an empirical dependence of the peak-to-peak width $\Delta B$ on the angle $\theta$ between the applied field and the
$z$-axis as $\Delta B = 0.86 + \sqrt{0.37\cos^2\theta + 0.12\sin^2\theta}$ (in mT), for both Lorentzian and Gaussian components.

**3. Theoretical background for orientation distribution analysis**

3.1. EPR spectrum of a powder sample with isotropic orientation distribution of the crystallites

The EPR of the $Cu^{2+}$ ion in cuprorivaite can be described by a spin Hamiltonian including only the Zeeman interaction with
the external applied magnetic field $\vec{B}$:

$$\widehat{H} = \mu_B \vec{B} \cdot \boldsymbol{g} \vec{S} \tag{1}$$

where $\mu_B$ is the electron Bohr magneton, $\boldsymbol{g}$ the electron g-factor matrix and $\vec{S}$ the electron spin. The hyperfine interaction with the central nucleus $^{63}Cu$ or $^{65}Cu$ is not to be considered in the Hamiltonian as it is averaged out by exchanged interaction between $Cu^{2+}$ ions (Binet, et al., 2021). The resonance field is then determined by the g-factor only and its anisotropy. An EPR spectrum is recorded by scanning the applied field at fixed frequency $\nu$ of the electromagnetic wave. For the magnetic field making an angle $\theta$ with the molecular z-axis, an EPR transition occurs at a resonance magnetic field

$$B_r(\theta) = \frac{h\nu}{\mu_B \sqrt{g_{\parallel}^2 \cos^2(\theta) + g_{\perp}^2 \sin^2(\theta)}} \tag{2}$$

with $h$ the Planck constant. The resonance field thus varies in a range defined by the boundary values $B_r^{\parallel} = B_r(\theta = 0) = h\nu/g_{\parallel}\mu_B$ and $B_r^{\perp} = B_r\left(\theta = \frac{\pi}{2}\right) = h\nu/g_{\perp}\mu_B$ with $B_r^{\parallel} < B_r^{\perp}$ in the present case. The EPR transition of a $Cu^{2+}$ ion thus depends on the orientation of the molecular frame $(x, y, z)$ within the laboratory frame $(X_0, Y_0, Z_0)$ defined such as $\vec{B} \parallel Z_0$, through the angle $\theta$. In the case of a powder sample where crystallites exhibit a perfectly isotropic distribution of orientations, the EPR spectrum is given by summing over all possible orientations $\theta$:

$$S(B) = \int_{\theta=0}^{\pi} \int_{\varphi=0}^{2\pi} \int_{\psi=0}^{2\pi} \omega(\varphi, \theta, \psi) f\left(B - B_r(\theta)\right) \sin\theta \, d\theta d\varphi d\psi \tag{3}$$

where $f\left(B - B_r(\theta)\right)$ is a normalized lineshape function, and $\omega(\varphi, \theta, \psi)$ is the transition probability, which depends on the orientation, given by the set of Euler angles $(\varphi, \theta, \psi)$, of the molecular frame within the laboratory frame. A typical EPR spectrum in the case of an isotropic distribution of crystallite orientations and for an axial g-matrix is shown in Fig. 2. It exhibits a low-field maximum at $B_r^{\parallel}$ and a baseline crossing close to (but not exactly at) $B_r^{\perp}$.

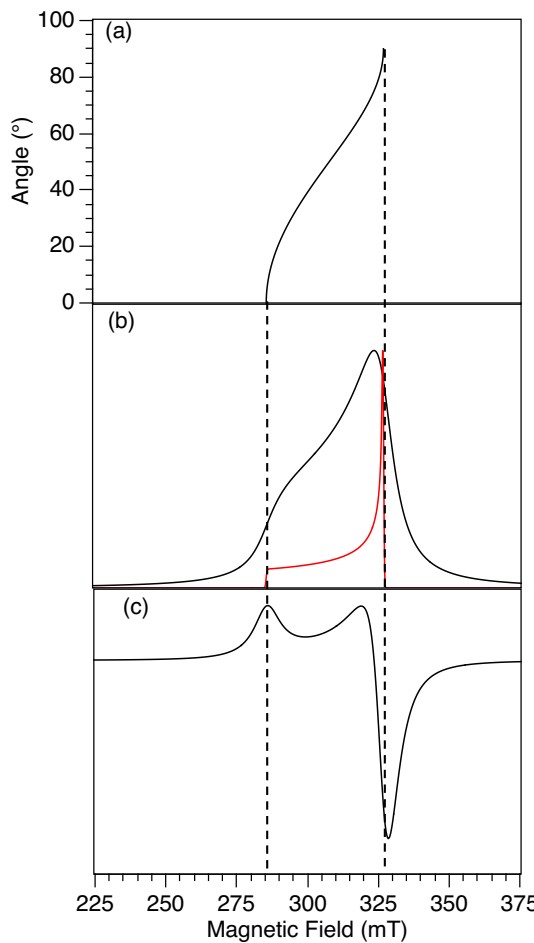

**Figure 2:** (a) Angular variation of the EPR resonance field according to Eq. 2. for $Cu^{2+}$ in cuprorivaite. (b) Spectral density in red and corresponding EPR absorption spectrum in black considering a non-zero linewidth. (c) Actual EPR spectrum corresponding to the absorption derivative.

3.2. Powder EPR spectrum with preferred orientation of the crystallites

To describe the EPR spectrum of a sample with a non-uniform distribution of crystallites orientations, three different frames need to be considered, namely the molecular frame $(x, y, z)$, the sample frame $(X, Y, Z)$ with $X$, $Y$ and $Z$ axes having a defined position with respect to the sample, and the laboratory frame $(X_0, Y_0, Z_0)$, as shown in Fig. 3a. The orientation of the molecular frame within the sample frame is specified by a set of three Euler angles $\Omega = (\alpha, \beta, \gamma)$, its orientation within the laboratory

frame by the set $\Omega' = (\varphi, \theta, \psi)$ and the orientation of the sample frame within the laboratory frame by the set $\Omega_0 = (\alpha_0, \beta_0, \gamma_0)$ (Fig. 3b).

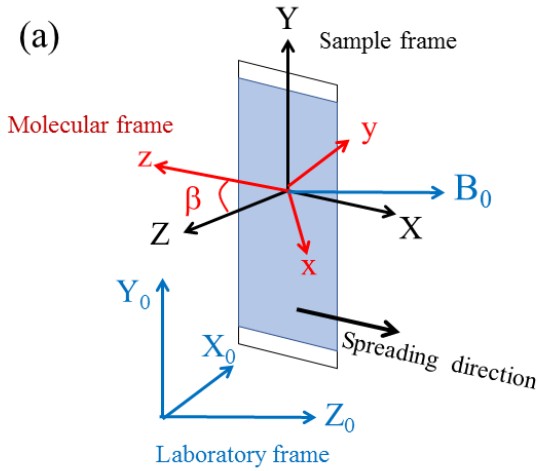

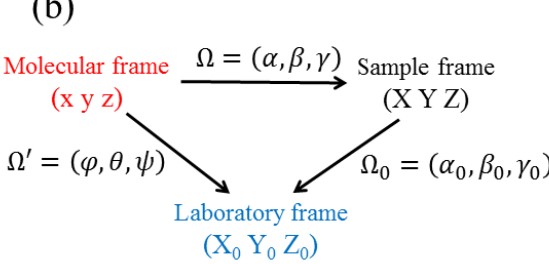

**Figure 3.** (a) Definition of the molecular, sample and laboratory frames. (b) Sets of Euler angles relating the various frames.

The relevant quantity to characterize the non-random orientations is the orientation probability density $\mathcal{P}(\Omega)$ of the crystallites in the sample frame $(X, Y, Z)$ such that $\mathcal{P}(\Omega)d\alpha \sin\beta \, d\beta d\gamma$ is the probability of the molecular frame orientation in the sample frame being found in the range $[\alpha, \alpha + d\alpha[ \times [\beta, \beta + d\beta[ \times [\gamma, \gamma + d\gamma[$. This density $\mathcal{P}(\Omega)$ is directly related to the way the pigment was applied on the sample. It is normalized so that $\int_{\alpha=0}^{2\pi} \int_{\beta=0}^{\pi} \int_{\gamma=0}^{2\pi} \mathcal{P}(\Omega)d\alpha \sin\beta \, d\beta d\gamma = 1$. For a specific orientation $\Omega_0 = (\alpha_0, \beta_0, \gamma_0)$ of the sample in the laboratory frame, the EPR spectrum is then given by:

$$S(B, \Omega_0) = \int_\Omega \mathcal{P}(\Omega)\omega(\Omega, \Omega_0)f\big(B - B_r(\Omega, \Omega_0)\big)d\Omega \qquad (4)$$

In the case of an isotropic orientation distribution, $\mathcal{P}(\Omega)$ is constant with $\mathcal{P}(\Omega) = 1/8\pi^2$ and can be omitted in Eq. 4, which then reduces to Eq. 3. The transition probability $\omega(\Omega, \Omega_0)$ scales as the $g$-value in the case of a field-swept EPR spectrum (Aasa and Vänngård, 1975). When the anisotropy of the g-factor is small, as is the case here, the transition probability is almost independent on the orientation $(\Omega, \Omega_0)$. It then appears as an irrelevant scaling factor and will be hereafter dropped for sake of

125 simplification. The determination of $\mathcal{P}(\Omega)$ from experimental EPR spectra is essentially adapted from Hentschel, et al. (1978). The unknown probability density $\mathcal{P}(\Omega)$ is first expanded as a combination of Wigner matrix elements $D_{mn}^{(l)}(\Omega)$:

$$\mathcal{P}(\Omega) = \sum_{l=0}^{\infty} \sum_{m=-l}^{l} \sum_{n=-l}^{l} p_{lmn} D_{mn}^{(l)}(\Omega) \tag{5}$$

where $D_{mn}^{(l)}(\alpha, \beta, \gamma) = e^{-im\alpha} e^{-in\gamma} \langle Y_{l,m} | \exp(-i\beta \hat{L}_y/\hbar) | Y_{l,n} \rangle$, $Y_{l,m}$ and $Y_{l,n}$ being spherical harmonics and $\hat{L}_y$ the $y$-component of the orbital angular momentum operator. An essential simplification can be introduced here because all samples happened to have a revolution symmetry about the $Z$-axis of the sample frame (see below) and because the g-factor is axially symmetric about the molecular $z$-axis (C4 axis) meaning that the spectrum is unchanged by any rotation of the crystallites about the molecular $z$-axis. The probability density $\mathcal{P}(\Omega)$ then no longer depends on $\alpha$ and $\gamma$ but only on $\beta$. The dependence on $\alpha$ and $\gamma$ arises from terms $p_{lmn} D_{mn}^{(l)}(\Omega)$ in Eq. 5 with non-zero values of $n$ or $m$, The general way to get rid of this dependence is to set $p_{lmn} = p_l \times \delta_{n,0} \times \delta_{m,0}$. This selects terms in Eq. 5 with $m = n = 0$ so that:

$$\mathcal{P}(\Omega) = \mathcal{P}(\beta) = \sum_{l=0}^{\infty} p_l D_{00}^{(l)}(\Omega) \tag{6}$$

where $D_{00}^{(l)}(\Omega) = P_l(\cos\beta)$ with $P_l(x)$ being the $l$-th order Legendre polynomial. The orientation probability density $\mathcal{P}(\Omega)$ is then fully determined by the set of coefficients $p_l$ with $l \in \mathbb{N}$, which must be obtained from experimental EPR spectra. Detailed calculations given in Supporting Information show that the EPR spectrum for a specific orientation $\Omega_0 = (\alpha_0, \beta_0, \gamma_0)$ of the sample in the laboratory frame is given by:

$$S(B, \Omega_0) = S(B, \beta_0) = \sum_{l=0}^{\infty} p_l P_l(\cos\beta_0) \int_{\theta=0}^{\pi} f(B - B_r(\theta)) P_l(\cos\theta) \sin\theta \, d\theta \tag{7}$$

From the above equation, the EPR spectrum only depends on the angle $\beta_0$ between the sample $Z$-axis and the applied magnetic field. Therefore, only a set of spectra at different $\beta_0$ angles are needed to determine the $p_l$ coefficients. The numerical implementation of the determination of the $p_l$ coefficients from a set of experimental spectra at different orientations $\beta_0$ of the sample in the laboratory frame is given in Supporting Information. It should be however noticed that since the EPR phenomenon is unchanged by reverting the orientation of the applied field, only $p_l$ coefficients with $l$ even can be determined so that the actual probability density $\mathcal{P}(\Omega)$ cannot be fully obtained from EPR.

**4. Orientation distribution of pigment crystallites in fluid oil under magnetic field**

The determination of the preferred orientation of pigment crystallites is illustrated in the case of cuprorivaite crystallites dispersed in a fluid medium, such as oil, and oriented by an external magnetic field. Under an external field, the magnetic potential energy per unit volume of a crystallite is

$$E(\theta) = -Ng(\theta)\mu_B B \langle S_z \rangle = -\frac{1}{2} Ng(\theta)\mu_B B \tanh\left(\frac{g(\theta)\mu_B B}{2kT}\right) \tag{8}$$

where $N$ is the number of $Cu^{2+}$ ions per unit volume, $\langle S_z \rangle$ the thermal average of the $Cu^{2+}$ electron spin component along the magnetic field direction. In a fluid, a pigment crystallite is free to rotate to minimize its potential energy. It appears from Eq.

8 that the potential energy is minimal for an orientation $\theta$ of the molecular $z$ axis with respect to the field corresponding to the maximum value of the $g$-factor $g(\theta = 0) = g_{\parallel}$. Therefore, we expect pigment crystallites to be mostly oriented so that their crystal $c$-axis (which is parallel to the molecular $z$-axis) is along the applied field, which here defines the sample $Z$-axis. After orientation in the field, the sample was cooled to freeze the oil and thus the grain orientations, and EPR spectra were recorded upon varying the angle $\beta_0$ between the measuring field ($Z_0$ direction) and the sample $Z$ axis. These spectra are shown in Fig.

4 for different values of $\beta_0$.

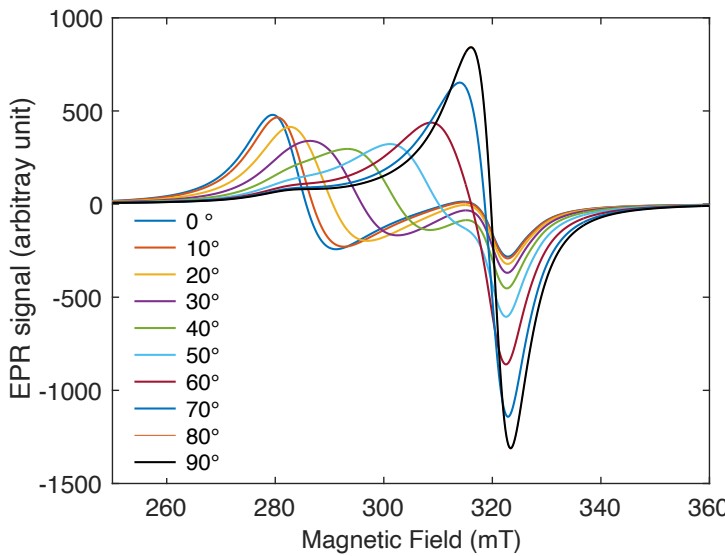

**Figure 4.** EPR spectra of frozen cuprorivaite grains magnetically oriented in oil for different angles $\beta_0 = (\widehat{Z_0, Z})$ between the orienting and the measuring fields. The rotation axis is $Y_0$ and the sample was set with $Y \parallel Y_0$.

For $\beta_0 = 0$ (measuring field parallel to the orienting field), the spectral density is mostly concentrated about the lowest resonance field $B_r^{\parallel} = \frac{h\nu}{g_{\parallel}\mu_B}$. When the angle $\beta_0$ increases up to $\frac{\pi}{2}$, the EPR spectral density shifts to higher fields, and is concentrated about the highest resonance field $B_r^{\perp} = h\nu/g_{\perp}\mu_B$ at $\beta_0 = \frac{\pi}{2}$ (measuring field perpendicular to the orienting field). This angular dependence of the EPR spectra shows that, as expected, crystallites have been predominantly oriented with their crystal $c$-axis along the orienting field. The orientation probability density $\mathcal{P}(\beta)$ calculated from the EPR spectra is shown in

Fig. 5a, where $\beta$ is also here the angle between the crystal $c$-axis and the orienting field. The calculated EPR spectra are given in Fig. S1. The polar representation of $\mathcal{P}(\beta)$ shows a large maximum of $\mathcal{P}(\beta)$ for $\beta = 0$ and can be compared to the probability density for an isotropic distribution of orientations (red circle in the middle of Fig. 5a). This clearly indicates a strongly anisotropic orientation distribution of the pigment grains. Figure 5b shows the $p_l$ coefficients in the expansion of $\mathcal{P}(\beta)$ in Eq. 6. Order $l = 0$ corresponds to the isotropic contribution to $\mathcal{P}(\beta)$ and values $l \neq 0$ to the anisotropic components. The highly

anisotropic character of $\mathcal{P}(\beta)$ is shown by the fact that the isotropic component $l = 0$ is not the dominant one but rather components with $l$ from 2 to 10.

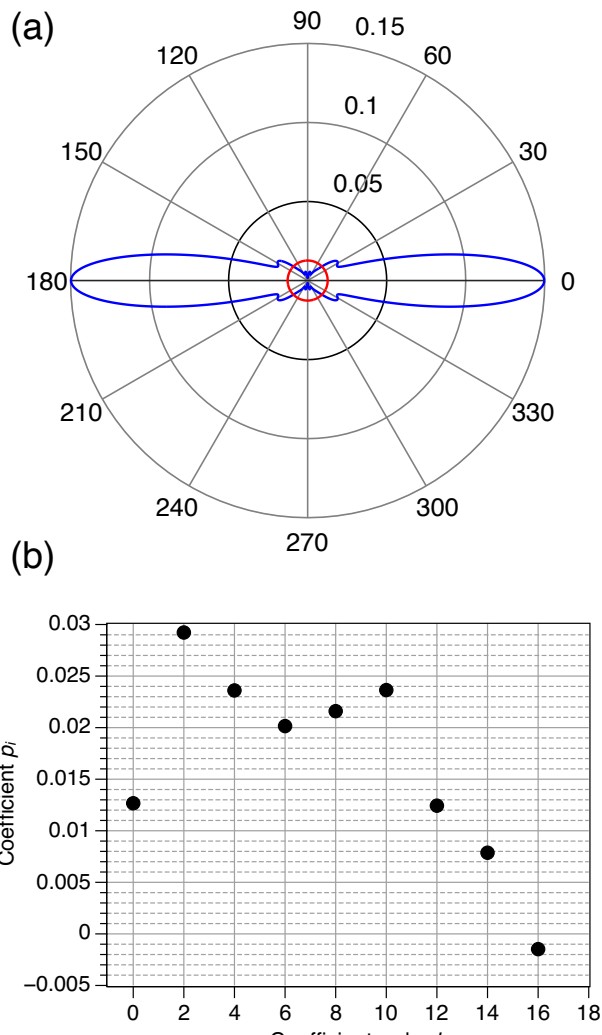

**Figure 5.** (a) Polar plot (in blue) of the orientation probability density $\mathcal{P}(\beta)$ for cuprorivaite magnetically oriented in fluid oil. The red circle represents a perfectly isotropic orientation distribution $\mathcal{P}(\beta) = 1/8\pi^2$. (b) Coefficients $p_l$ in the expansion of

$\mathcal{P}(\beta)$ according to Eq. 6. Odd-order coefficients are all zero.

## 5. Orientation distribution of pigment grains in dried films: effect of the application process

5.1. Films deposited with the applicator

Films were deposited on a flat substrate with an applicator by a one-way spreading along the $X$-direction in the substrate plane

(Fig. 3a) of a mixture of fluid linseed oil and pigment grains. This application process is expected to be the most reproducible

one among those used in this work. Since an anisotropy along the *X*-direction is induced by the process, it is meaningful to determine if a preferred orientation of the grains is induced and how the distribution is oriented with respect to the spreading *X*-direction. EPR spectra corresponding to the rotation of the sample about the sample *Z*-axis and about the sample *Y*-axis are shown in Figs. 6a and 6b, respectively. It turns out that there is no significant change in the spectrum shape upon rotation about

the *Z*-axis (Fig. 6a). This means that the orientation distribution has a revolution symmetry about the *Z*-axis.

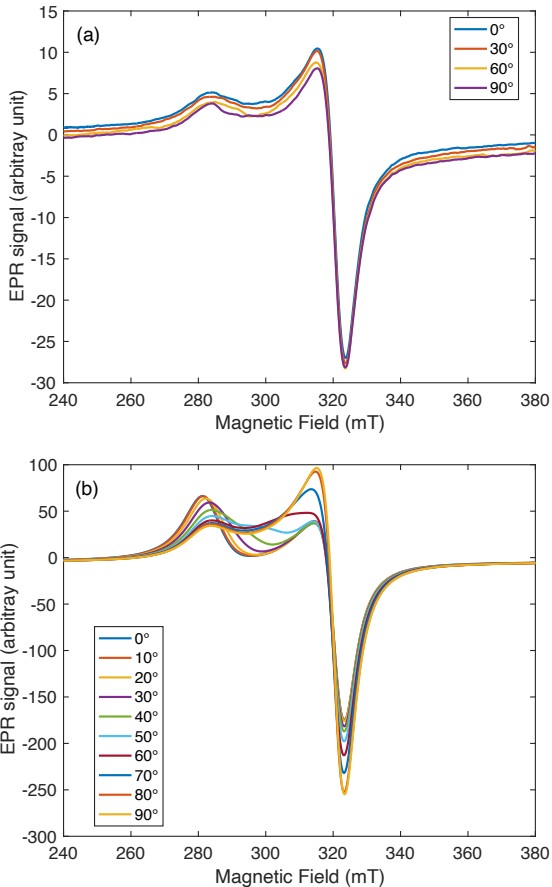

**Figure 6.** EPR spectra of a film deposited with an applicator upon rotation about (a) the *Z*-axis, (b) the *Y*-axis, at different angles $\beta_0$ between the magnetic field and (a) the *X*-axis or (b) the Z-axis. The rotation axis is $Y_0$. The sample was set with $Z \parallel Y_0$ in case (a) and with $Y \parallel Y_0$ in case (b).

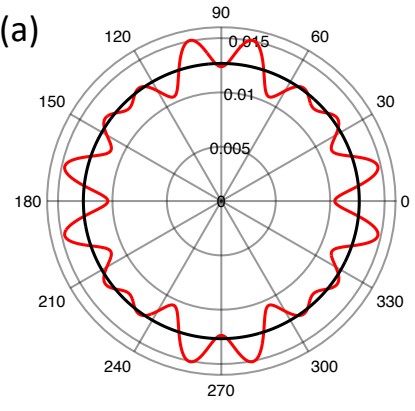

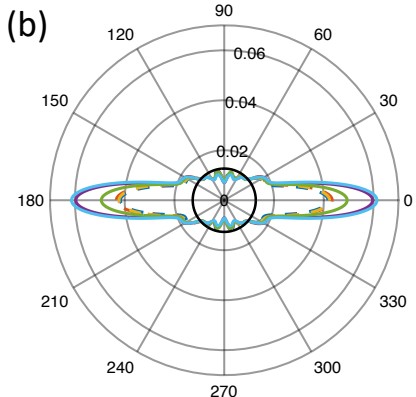

**Figure 7.** Orientation probability densities $\mathcal{P}(\beta)$ for 6 films deposited under same conditions with an applicator calculated from (a) a rotation of the sample about the $Z$-axis ($\beta$ being the angle from the $X$-axis in the $XY$-plane), (b) a rotation about the $Y$-axis ($\beta$ being the angle from the $Z$-axis in the $XZ$-plane). The black circles correspond to an isotropic orientation.

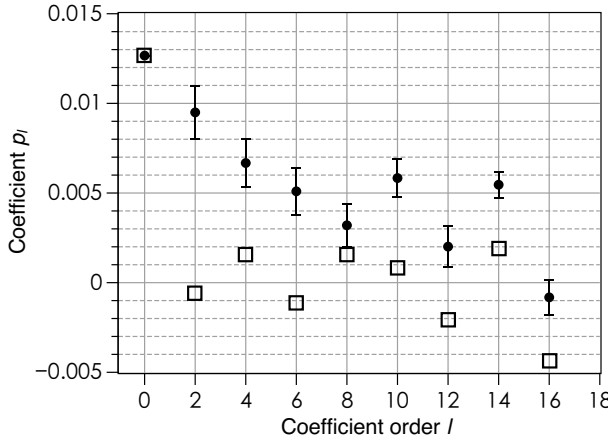

**Figure 8.** Coefficients $p_l$ in the expansion of $\mathcal{P}(\beta)$ according to Eq. 6 for a rotation about the $Z$-axis (open squares) and a rotation about the $Y$-axis (black circles) in the case of films deposited with an applicator. Odd-order coefficients are all zero.

This is confirmed by the polar plot of the corresponding density $\mathcal{P}(\beta)$ (here $\beta$ is the angle from the $X$-axis in the $XY$-plane),
which very close to a circle (Fig. 7a) and by the $p_l$ coefficients close to zero except for $l = 0$ (Fig. 8). In this figure, the values
of $p_l$ coefficients corresponding to an isotropic distribution of crystallite orientations are represented by a discontinuous red
line.

Significant changes in the EPR spectrum shape are observed upon rotation about the $Y$-axis, the magnetic field direction
moving from the $Z$-direction ($\beta_0 = 0$) perpendicular to the sample plane to the $X$-direction ($\beta_0 = \pi/2$) in the sample plane
(Fig. 6b). The low field part of the spectrum, corresponding to crystallites with the $c$-axis parallel to the field is enhanced at
$\beta_0 \approx 0$, while the high field part (crystallites with the $c$-axis perpendicular to the field) is enhanced at $\beta_0 \approx \pi/2$. This means
that crystallites are mostly oriented with their $c$-axis perpendicular to the sample plane. This is confirmed by the probability
density $\mathcal{P}(\beta)$ being maximum at $\beta = 0$ (Fig. 7b). Calculated EPR spectra are given in Figs. S2 and S3. The variability in the
preferred orientation of the crystallites has been analyzed in a set of six samples prepared under the same conditions and the
corresponding $\mathcal{P}(\beta)$ densities shown in Figure 7b with different colors. The average values of the $p_l$ coefficients over those
six samples are plotted in Figure 8 with the error bars given by their standard deviation. The maximum value $\mathcal{P}(\beta = 0) =$
$0.04 - 0.06$ of the probability density for the films is lower than in the case of magnetically oriented grains in fluid oil where
the maximum value was $\mathcal{P}(\beta = 0) \approx 0.15$. For films, the highest value of $p_l$ is at $l = 0$ corresponding to the isotropic
contribution to $\mathcal{P}(\beta)$, while for magnetically oriented grains in fluid oil the anisotropic contributions at non-zero orders $l =$
$2 - 10$ dominate the isotropic one (Fig. 5b). All these results indicate that the preferred grain orientation in dried films after
mechanical application is less pronounced than under an external magnetic field in fluid oil.

5.2. Preferred orientations with manual deposition modes

The preferred orientation with more operator-dependent modes of deposition were also investigated, namely spreading along
the $X$-direction in the sample plane with a paintbrush, dabbing onto the substrate with a paintbrush and droplet deposition with
a pipette. In each case two or three samples were prepared and analyzed. All samples exhibited a revolution symmetry about
the $Z$-axis as in the case of films deposited with the applicator. The EPR spectra under rotation about the $Y$-axis are given in
Figs S4 to S6 in the Supplementary Information. The orientation probability densities are shown in Figures 9a and 9b and the
corresponding $p_l$ coefficients (average values over 2-3 samples and standard deviations as error bars) in Fig. 10.

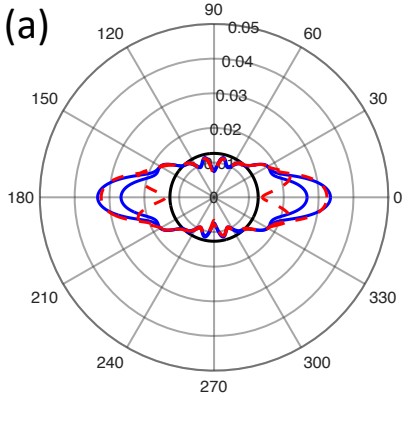

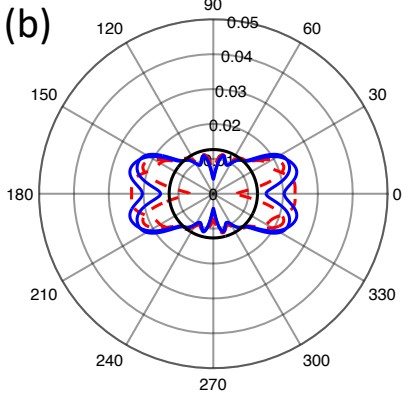

**Figure 9.** Orientation probability densities $\mathcal{P}(\beta)$ for films made by (a) brush-painting (full blue curves) and strong brush-painting (dashed red curves), (b) dabbing (full blue curves) and droplet deposition (dashed red curves). The black circles correspond to an isotropic orientation distribution. The sample setting in the laboratory frame was identical to the case of Fig. 6b.

In all cases, the probability densities $\mathcal{P}(\beta)$ exhibit a maximum value for $\beta$ in the range $0-17°$. This again shows a preferred orientation with the crystal $c$-axis close to the sample $Z$-axis. However, the maximum values of $\mathcal{P}(\beta)$, about $0.02-0.03$ are significantly lower than for films deposited with the applicator. This is in line with the $p_l$ coefficients for $l \neq 0$ in Fig. 10, which are lower than for films deposited with the applicator showing that the orientation distribution with paint-brushing,

 dabbing or droplet deposition is less anisotropic than when the film is deposited with an applicator. When using a paintbrush, increasing the strength of the gesture ("Strong brush-painting" data as compared to "Brush painting" data in Figs. 9a and 10) does not seem to induce a significant difference in the orientation distribution, as shown by close $\mathcal{P}(\beta)$ curves in Fig. 9a and almost equal leading $p_l$ coefficients for $l = 2,4,6$ in Fig. 10. The orientation distributions in the cases of dabbing and droplet

deposition appear to be slightly less anisotropic than for brush-painting for their maximal values of $\mathcal{P}(\beta)$ are closer to the lower bound 0.02 and for they have lower values of the leading $p_l$ coefficients for $l = 2,4,6$.

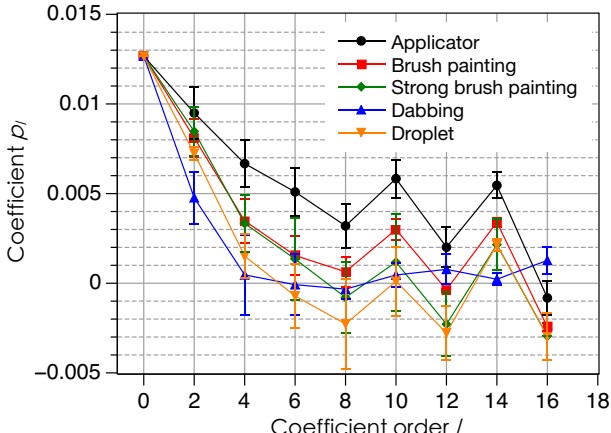

**Figure 10.** Coefficients $p_l$ in the expansion of $\mathcal{P}(\beta)$ according to Eq. 6 for the various deposition modes of films. The discontinuous red line corresponds to an isotropic distribution of orientations. The continuous lines are only guides for the eye and odd-order coefficient are all zero.

5.3. Possible cause of the preferred orientation

Whatever the application mode of the oil-pigment mixture, two common features were revealed by the above analyzes: (*i*) the pigment crystallites exhibit a preferred orientation with their crystal *c*-axes oriented almost perpendicularly to the sample plane, (*ii*) the orientation distribution has a revolution symmetry about the normal to the sample plane. More surprising is that these features are observed even when a mechanical stress is applied along the sample plane (*X*-axis) during the deposition (whether with the applicator or with a paintbrush) and that this mechanical stress seems to induce the most pronounced preferred orientations. It must also be mentioned that during the film deposition and drying, the gravitational force was perpendicular to the sample plane. A possible explanation for this preferred orientation is based on the sedimentation of the pigment crystallites within the oil while the latter is still viscous. Indeed, microcrystals of cuprorivaite exhibit a tabular morphology (Bloise, et al., 2016). Then when microcrystals fall through the liquid oil under the gravitational force, they tend to lay down with their largest face horizontal (Fig. 11a). Consequently, microcrystals adopt a preferred orientation with the normal to the largest face perpendicular to the sample plane. The piling of the crystals is not perfect, some crystals being tilted or aggregated in larger grains. This is mostly the case of the droplet or dabbing deposition. When a horizontal mechanical stress is applied with an applicator or a paint brush, the polymer chains of the oil get aligned along the stress direction (Fig. 11b). This may have two effects. First a dissociation of the aggregates into separate microcrystals, second a better alignment of the crystal tablets in the horizontal plane by polymer-crystal surface interactions. This could explain why the preferred orientation seems to be enhanced when such a mechanical stress is applied during film deposition.

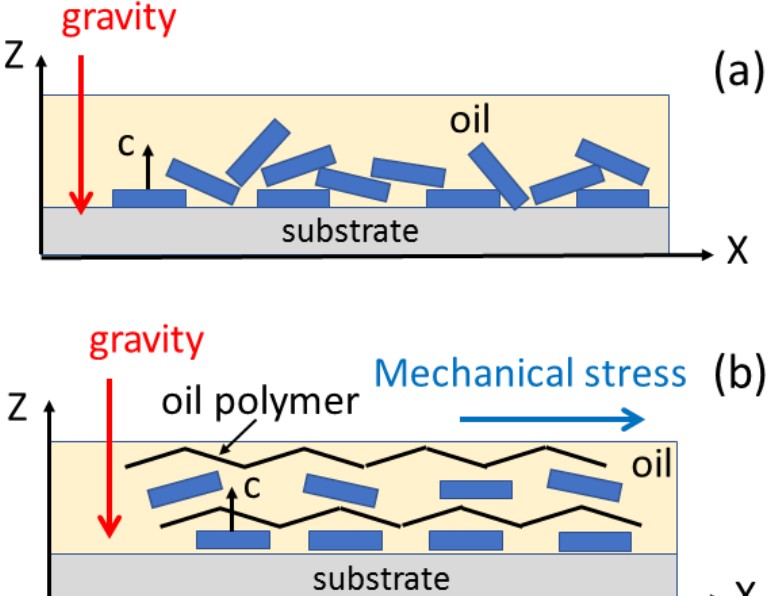

**Figure 11.** Models for the sedimentation of pigment microcrystals in oil leading to (a) poorly oriented crystals upon dabbing or droplet deposition, (b) an enhanced orientation when the oil polymers are stretched by an in-plane mechanical stress in the case of brush-painting or deposition with an applicator.

## 6. Can EPR reveals the gesture of an artist?

We have seen that three competing forces can contribute to the preferential orientation of pigment grains in an organic binder: a magnetic field, gravity, and a mechanical stress. The orientation in a magnetic field could be potentially interesting if it is effective in the Earth field. However, in the present work on Egyptian blue pigment, we could hardly observe the orientation effect in a field weaker than 20 mT, which is 1000 times higher than the earth field. The effect of gravity and mechanical strain can induce a preferential orientation only if pigment grains have a symmetry lower than cubic, *i.e.* in the case of platelets, cylinders, rods etc... In the case of Egyptian blue, whose crystallites are in the form of platelets, these two forces orientate the platelets parallel to each other. Therefore, if the substrate is horizontal, gravity and mechanical stress will both orient the platelets parallel to the substrate, as shown in the experiments described in this work (Fig.11). Nonetheless, these two situations can be distinguished by a careful analysis of the EPR lineshape, as shown above. However, the platelet-shape of the crystallites does not allow to identify the direction of the painter's gesture because the *z* axis of the platelets is perpendicular to the spreading gesture, whatever the direction of this movement. In principle, the frescoes were painted on walls, and therefore on vertical supports, whereas the pigment was deposited on horizontal supports in the present work. Unfortunately, it was not

possible to test experimentally the effect of verticality because the binding-oil used in this work is a viscous liquid that dries very slowly, so that it flows down when the sample is placed vertically.

The ideal case for determining artist's gesture by EPR would certainly be the case where pigment crystallites have the shape of elongated parallelepipeds, rods, cylinders, with an additional condition that the *z*-axis of the molecular frame is along the main axis of the crystallite. In the case of a horizontal substrate, it is anticipated that gravity will randomly orient the *z*-axes of the rods in the XY plane of the sample frame, while application of the pigment will orient the rods in the direction of the gesture. One would then expect to obtain orientation probability densities that would precisely reflect the artist's intentions.

An important point to consider is the applicability of the method. With traditional EPR, which uses a closed resonant cavity containing a small diameter tube, only small fragments of fresco could be analyzed. This situation often occurs in archaeology, where fragments have spontaneously become detached from walls. In some cases, and only with permission, it is possible to sample fragments from frescoes or paintings that can are sufficiently small to fit into EPR tubes. The ideal situation, towards which we should move, is the possibility to study an entire object *in situ* and non-invasively. A low frequency (355 MHz) EPR spectrometer for detecting and imaging paramagnetic species on a flat surface has been recently developed by Hornak's team (Switala, et al., 2017). However, at such low field, the anisotropy of g-factor cannot be resolved so that the orientation effects described in the present work cannot be observed. In the project that finances this work (https://anr.fr/Project-ANR-17-CE29-0002), it is planned to build a portable EPR spectrometer, working at 5 GHz, based on planar micro-resonators of the "microstrip" type. In this case, planar (or nearly planar) and large objects, which is the case of frescoes and paintings, can be analyzed by moving and rotating the spectrometer over the surface of the object.

**6. Conclusion**

 This work shows that the simple act of spreading a paramagnetic pigment dispersed in a binder on a surface introduces a preferential distribution of grains orientations on the paint layer that can be detected by EPR. This effect has been demonstrated with Egyptian blue, a pigment that was used for several millennia in antiquity. By analyzing the variation of the EPR line shape as a function of the angle between the magnetic field and the surface of the layer, the orientation probability density of the grains can be determined and was found to deviate significantly from the isotropic distribution. It is possible to differentiate between orientation distributions induced by mechanical spreading and those in which only gravity is at work (droplet deposition, dabbing). This effect has been demonstrated with a pigment (Egyptian blue) whose grains are in the form of platelets. It would certainly be greater for rod-shaped grains.

**Author Contributions.**

Elise Garel prepared the samples and performed the EPR experiments. Elise Garel and Laurent Binet processed the data and performed the calculations. The three authors discussed and interpreted the results. Laurent Binet and Didier Gourier wrote the paper. All authors have read and agreed to this version of the manuscript.

**Competing interests.** The authors declare no competing financial interest.

**Acknowledgment.**

**Financial support.** This work received funding from *Agence Nationale de la Recherche* (ANR) under contract N° ANR-17-CE29-0002-01.

**Supplementary information.** Derivation of Eq.7; Numerical calculation of the coefficients $p_l$ from experimental EPR spectra; Matlab script for the calculation of the orientation probability density; Experimental and calculated EPR spectra;

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
