# Peer review of "Mechanical ordering of pigment crystallites in oil binder: Can EPR reveal the gesture of an artist?"

_Magnetic Resonance, 2022_

## Author Comment (AC1)

**Point-by-point reply to referee # 1 for the manuscript:**

*Mechanical ordering of pigment crystallites in oil binder: Can EPR reveal the gesture of an artist?,* by E. Garel, L. Binet, and D. Gourier.

We sincerely thank the reviewer for his detailed and critical reading of the manuscript, particularly the theoretical part, which also allow us to address some points that need to be clarified.

**Comment #1:** *The discussion mentions that the crystallites are platelets. Authors should provide experimental evidence that the crystallites in their samples preparations are indeed platelets. There is a reference for this (Bloise, 2016), but the preparation procedure in that cited paper appears somewhat different.*

**Response:** It is important to note that the shapes of crystallites are mainly determined by the crystallographic structure of the mineral. When the structure is composed of covalent two-dimensional sheets linked by weaker interactions, the crystals are always lamellar. This is the case of cuprorivaite, which belongs to the group of phyllosilicates, all of whose minerals have the form of platelets (this is the case of micas, clay minerals etc.). The structure of cuprorivaite is made up of double layers of corner-sharing [SiO$_4$] tetrahedra, separated by a layer of Ca$^{2+}$ ions. It is this two-dimensional structure that dictates the platelet shape of the crystallites, regardless of the route of synthesis (natural or artificial).
A paragraph has been added to clarify this point (lines 45 to 49, in red).

**Comment #2:** *It is stated that the g-parallel axis is along the crystal c axis and that the c axis is normal to the platelet plane. Evidence for this should be presented, or cited, since it is essential for the interpretation of the data.*

**Response:** This is an important point, thank you. The structure of cuprorivaite is tetragonal, with space group P4/ncc, which means that the crystallographic c axis (C4 axis) is normal to the silicate layers, and thus to the platelets. Cu$^{2+}$ ions occupy octahedral sites with D$_{4h}$ point symmetry, with the C4 axis (which is also the g-parallel axis) parallel to the crystallographic c-axis, and thus normal to the platelet plane.
A sentence has been added to show that g// is normal to the platelets (lines 48- 50, in red).

**Comment #3:** *In the discussion, it is hypothesized that the orientational distribution is affected by gravity and depends on whether the sample plane is horizontal or vertical during film drying. Only the horizontal case is shown experimentally. Why not the vertical? This would support the discussion that suggests that the platelets would then orient differently. Without experimental data on vertically dried samples, the discussion about the role of gravity is pure speculation and not useful.*

**Response:** We agree that the discussion on the effect of gravity on the orientation distribution is speculative regarding the effect of the vertical position of the sample. It was not possible to test this effect experimentally because binding-oil is a viscous liquid that dries very slowly. This liquid flows down when the sample is placed vertically. A sentence has been added in Section 5 to clarify this point (lines 288-289, in red).

**Comment #4:** *How are the 25x2-3 mm sample strips rotated around the axis perpendicular to the sample plane? This cannot simply be achieved by putting the sample strip into an EPR tube and rotating the EPR tube around its axis which is along the lab Y0 axis (Fig.3) in a standard EPR spectrometer. The experimental setup should be described in more detail.*

**Response:**
In that case, samples with size 2 x 2 mm² that fit into standard sample tubes were used. This has been added in the experimental section (lines 59-60, in red).

**Comment #5:** *Were all samples dried with the application plane horizontal? This should be clarified in the Experimental section.*

**Response:**
Yes. This has been clarified in the experimental section (lines 52-53, in red).

**Comment #6:** *To fully descibe the experimental setup, it is not sufficient to specify just the rotation axis relative to the sample (e.g, Z and Y in Fig.6), but rather (i) the orientation of the sample in the spectrometer (lab frame), and (ii) the rotation axis in the lab frame. Only then is the description complete.*

**Response:**
The rotation axis in the laboratory frame was the $Y_0$ axis. The sample setting in the laboratory frame has been specified in the captions of Figures 4, 6 and 9 (in red).

**Comment #7:** *In general, it is more correct to refer to "random orientation distribution" as "uniform orientational distribution" or "isotropic orientational distribution". In analogy, a "non-random distribution" is better referred to as "non-uniform". This applies to several places in the manuscript.*

**Response:**
We agree with the reviewer's comment and the text has been corrected accordingly.

**Comment #8:** *The reduction from the triple sum in Eq.(5) to the single sum in Eq.(6) should be shown mathematically, to improve clarity and rigor. In my understanding, one angle can be dropped because of the observed rotational symmetry of the EPR spectrum when the sample rotated around the Z axis. But what is the reason the second angle is dropped? The axial symmetry of the EPR spectrum?*

**Response:**
The reviewer is right. The fact that the probability density $\mathcal{P}(\Omega)$ does not depend on angles $\alpha$ and $\gamma$ is a consequence of *both* the rotational symmetry about the sample Z-axis *and* the axial symmetry of the g-factor (spectrum independent on any rotation of a crystallite about the molecular z-axis). Since the general expansion $\mathcal{P}(\Omega) = \sum_{l=0}^{\infty} \sum_{m=-l}^{l} \sum_{n=-l}^{l} p_{lmn} D_{mn}^{(l)}(\Omega)$ (Eq. 5) is a linear combination of functions of angles $(\alpha, \beta, \gamma)$ where the dependence on $\alpha$ and $\gamma$ arises from terms with non-zero values of $n$ or $m$, the only way to get rid of the $\alpha$ and $\gamma$ dependence is to set $p_{lmn} = p_l \times \delta_{n,0} \times \delta_{m,0}$. This selects only terms with $n = m = 0$ in Eq. 5, thus yielding Eq. 6.
An explanatory sentence has been added to the text (lines 130-134, in red).

**Comment #9:** *Eq.(3) is only approximately correct, since the transition probability depends on the g-factor along the microwave B1 field - for this, all three Euler angles are in principle needed. See e.g. the textbook by J.R.Pilbrow. The approximation inherent in Eq.(3) should be explicitly stated.*

**Response:**
The reviewer is right again, and correction is made in Eq. 3. However, when the g-factor anisotropy is very weak as is the case here, $\omega(\varphi, \theta, \psi)$ is almost constant and can be dropped. More specifically, it is dropped from Eq. 7 to justify the integration over $\theta$ only, in the right-hand member.

**Comment #10:** *Spectra simulations for the case in Fig.6(a) are missing from the SI.*

**Response:**
These spectra were added in the SI (new figure S2).

**Comment #11:** *What is the reason the spectra in Fig.6a are more noisy than all the other experimental spectra presented in the paper?*

**Response:**
It is just because in that case the sample was smaller ($2\times2$ mm$^2$) than other samples ($25\times2$ mm$^2$).

**Comment #12:** *The lines in Fig.5(b) and Fig.10 are misleading, as they suggest non-zero contributions for odd l. I suggest to remove these lines and present the data in these figures as is done in Fig.8.*

**Response:**
The lines in Figure 5b has been removed. We however prefer keeping the lines in Figure 10 for sake of clarity, given the large number of intermixed data points. We specified that these lines are only guides for the eye and that the odd-order coefficients are zero (lines 248-249, in red).

**Comment #13:** *In Figs. 8 and 10, the red dashed line should be removed. It suggests a continuous function, whereas the x axis is discrete. In this context, the tick marks in Fig.10 every 0.4 on the x axis make no sense.*

**Response:**
The figures were corrected according to the reviewer's request.

**Comment #14:** *The left-hand side of Eq.(S3) is missing the omega(Omega') factor.*

**Response:**
Eq. S3 was corrected. In addition, a comment was added in the SI (in red) justifying why $\omega(\Omega')$ was dropped in the following equations on the basis that $\omega(\Omega')$ is almost independent of $\Omega'$.

**Comment #15:** *What happened to the first integral in (S3)? Shouldn't there be an additional 2*pi factor for this integral appearing in (S5)?*

**Response:**
True. Equation S5 has been corrected accordingly.

**Comment #16:** *Figure S1 should include the simulation for all spectra in Fig. 4, not only a subset. Same applies to Figs. S2-S4.*

**Response:**
The requested simulations were added, see Fig. S1.

**Comment #17:** *It should be stated somewhere that the odd-integer components of the orientational distribution give an EPR spectrum that appears isotropic. Therefore, as far as I understand it, odd l cannot be distinguished from l=0.*

**Response:**

The odd-integer components do not yield an isotropic EPR spectrum. As an example, we show below theoretical spectra calculated from Eq. S6 in the case $p_0 = 1$ and $p_l = 0$ for $l > 0$ ("l = 0 only, isotropic case") and in the case $p_1 = 1$ and $p_l = 0$ for $l \neq 1$ ("l = 1 only").

[Figure]

The EPR spectra are clearly different.

**Comment #18:** *- Line 113: matrice -> matrix*

*- The surname of the last author in the Hentschel reference is Spiess, not Speiss.*

*- The first line in SI section 2 should refer to Eq. S6, not Eq. S7.*

*- Eq.(S2) should say p_l and not p_{l00}, to be consistent with other equations.*

*- After Eq.(S2), it should say "determines an orientation Omega'" (prime is missing)*

**Response:**

All these corrections have been made.

---

## Author Comment (AC2)

**Point-by-point reply to referee # 2 for the manuscript:**

*Mechanical ordering of pigment crystallites in oil binder: Can EPR reveal the gesture of an artist?,* by E. Garel, L. Binet, and D. Gourier.

We sincerely thank the reviewer for his detailed and critical reading of the manuscript, and in particular for raising two important points.

**Main comment:** *The work is performed accurately, and the results are, as whole, sound. The subject is of interest for Magn. Reason. Discussion, and is a nice tutorial on the effect of preferential orientation on EPR spectra. However I have a major point to raise as to the general applicability of the method. Indeed, while authors performed their measurements on samples prepared on purpose, the application of the proposed method on real samples would require detachment of part of the painting to study. Does not this risk to be seen as a destructive method? A comment on this should be added in the Introduction.*

**Response:** This is indeed an important issue, and we thank the reviewer for this comment. With traditional EPR, which uses a closed resonant cavity containing a small diameter tube, only small fragments of fresco could be analysed. However this is a situation that often occurs in archaeology when analysing ancient sites, where fragments of frescoes have spontaneously become detached. In some cases, and only with permission, it is possible to take small samples from a fresco. The ideal situation, towards which we should move, is the possibility to study an entire fresco *in situ* and non-invasively. In the project that finances this work, it is planned to build a portable EPR spectrometer, working at 5 GHz, based on planar resonators of the "microstrip" type. In this case, planar (or nearly planar) objects, which is the case of frescoes, can be analysed by moving the spectrometer over the surface of the object. This equipment is currently under construction (https://anr.fr/Project-ANR-17-CE29-0002 ).
A paragraph clarifying this point has been added (in red) at the end of section 6 (lines 294-304).

**Comment #1:** *Sample preparation: How were powders ground? Does the grinding procedure affect the spectra? Can authors provide an estimate of the size and anisotropy of the microcrystals they used (e.g. by the analysis of PXRD spectra)? I can imagine that the aspect ratio of the microcrystals can be modified by grinding, and this should result in different probability density as a function of the angle. This may have major consequences for the proposed method, and has to be considered.*

**Response:** We did not observe any effect of grinding on the EPR spectra, at least in the series of experiments we performed. For example, the spectra obtained in Binet et al. 2021, carried out on raw synthetic powders, and those obtained in this work, where the powders are ground to be dispersed in a binder, are identical. The grain sizes are widely dispersed and range mainly from 1 to 20 micrometers. These sizes are too large to produce Scherrer broadening of diffraction peaks in PXRD, which would not be the case if the sizes were of the order of a few hundred nanometres or less.

**Comment #2:** *Linewidth: can authors think of extracting some further information from the empirical dependence of the linewidth they extracted from the analysis of their spectra? Or, seen from a different point of view: is there a reason for the chosen functional dependence of the linewidth?*

**Response:** The lineshape is determined by weak antiferromagnetic interactions between neighbouring $Cu^{2+}$ ions, which average out the hyperfine interaction with $^{63}Cu$ and $^{65}Cu$ nuclei Binet et al. 2021). Due to the two-dimensional character of the cuprorivaite structure, the antiferromagnetic interaction is itself anisotropic, respecting the tetragonal symmetry of the copper sites. This is responsible for the angular variation of the line width. This anisotropy is, however, quite small (1.47 mT and 1.20 mt for the parallel and perpendicular linewidths, respectively), and therefore does not provide any new information on $Cu^{2+}$ ions and their mutual interactions. For this reason we have chosen an empirical expression for the angular variation of the line width.

**Comment #3:** *It looks like the simulation of the spectra is much better when the orientational distribution is more isotropic: indeed, the simulated spectra for the sample magnetically oriented in fluid oil are much less convincing than the remaining ones. Authors should provide a rationale for this.*

**Response:**
A potential and straightforward explanation would be the truncation of the expansion Eq. 6 of $\mathcal{P}(\Omega)$ for numerical implementation at too a small order. The calculated spectra shown in Fig. S1 were obtained with a truncation at l=16. We checked that adding more terms in Eq. 6, up to l=24, did not improve the simulation at all, so that this explanation must be ruled out. Unfortunately, we could not figure out any explanation why the simulations in the case of the magnetically oriented sample do not match as much the experimental spectra as in the other cases.

**Comment #4:** *The description of the different frames is somehow unclear: they state "the sample frame (X, Y,Z) with X, Y and Z axes having a defined position with respect to the sample." If I got it correctly from Figure 3, they mean that the sample frame (X,Y,Z) define the orientation of the sample with respect to the laboratory reference frame. This has to be clarified.*

**Response:**
We agree that the connection between the different frames can be confusing at some points. Indeed, the rotation axis in the laboratory frame was the $Y_0$ axis. For rotations about the sample Y-axis (magnetic field moving from the sample Z-axis to the sample X-axis), the sample Y-axis was set parallel to the laboratory $Y_0$ axis. This was the case of spectra or data in Figs. 4, 6b, 9 and 10. For rotations of about the sample Z-axis (Fig 6a), the latter was set parallel the laboratory $Y_0$ axis. These setting are now specified (in red) in the captions of the corresponding figures.

**Comment #5:** *Did authors try to use larger magnetic field to orient the fluid dispersion? It appears to me that 20 mT is quite a low field, and higher fields should result in even more pronounced preferential orientation.*

**Response:** Of course we used stronger magnetic fields (at least ten times stronger) for the field orientation, but the orientation distributions were the same. What would have been interesting is to obtain orientation effects under very weak magnetic fields (Earth field). But we have not observed anything for fields of less than about 10 mT.

**Comment #6:** *The sentence "It must also be mentioned that during the film deposition and drying, the Z-axis is along the vertical direction so that the gravitational force is perpendicular to the sample*

*plane" is a bit misleading: it is not a matter of the orientation of the Z axis (which is a choice of the authors), but rather of the fact that the gravitational force is perpendicular to the sample plane.*

**Response:**
The reviewer is right. It is actually a matter of gravity being perpendicular to the sample plane during drying. The misleading phrase "the $Z$-axis is along the vertical direction" was removed.

**Comment #7:** *Authors state that, in case of a vertical substrate, "gravity should tend to orient the plates perpendicular to the substrate": it is however hard to see how this could happen, given the platelet form of the pigment. I can imagine an accumulation of the pigment on the low end of the substrate, but I cannot see why they should pile differently. A more detailed explanation of the reason for the expected effect should be given.*

**Response:** We fully agree with this remark. This discussion about the effect of vertical position of the substrate is speculative and difficult to test experimentally, because in our experiments the oil containing the pigment is a viscous liquid, so the paint layer will flow along the surface if the support is placed vertically. We have therefore removed this discussion on the verticality of the substrate in Section 6. This discussion is now replaced by a sentence (lines 285-288, in red).